

# Cross-tolerance evolution is driven by selection on heat tolerance in *Drosophila subobscura*

Luis E. Castañeda[1,2]

[1] Núcleo Interdisciplinario de Biología y Genética, Instituto de Ciencias Biomédicas (ICBM), Facultad de Medicina, Universidad de Chile, Santiago, Chile
[2] Research Ring in Pest Insects and Climate Change (PIC2), Santiago, Chile

## ABSTRACT

The evolution of heat tolerance is crucial for the adaptive response to global warming. However, it depends on the genetic variation present in populations and the intensity of thermal stress in nature. Experimental selection studies have provided valuable insights into the evolution of heat tolerance. However, the impact of the heat stress intensity on the correlated changes in resistance traits under selection of heat tolerance has not yet been explored. In this study, the correlated response of increasing knockdown temperature in *Drosophila subobscura* was evaluated on the knockdown time at different stressful temperatures, the thermal death time (TDT) curves, and the desiccation and starvation resistance. Selection for increased heat tolerance was conducted using different ramping temperatures to compare the effect of heat intensity selection on resistance traits. An evolutionary increase of high temperature tolerance also confers the ability to tolerate other stresses such as desiccation and starvation. However, the extent to which these correlated responses depend on the intensity of thermal selection and sex may limit our ability to generalize these results to natural scenarios. Importantly, this study confirms the value of the experimental evolutionary approach in exploring and understanding the adaptive responses to global warming.

## INTRODUCTION

Rising environmental temperatures are a major challenge for ectotherms (*i.e.*, organisms whose body temperature depends on the ambient temperature) because their morphology, physiology, behavior, and performance depend on the thermal environment (*Huey & Stevenson, 1979*; *Cossins & Bowler, 1987*; *Angilletta, 2009*). Furthermore, rising environmental temperatures increase the risk of extinction for many species living near their upper thermal limits (*Deutsch et al., 2008*; *Huey et al., 2009*; *Hoffmann & Sgrò, 2011*). However, ectotherms can avoid the negative effects of heat through behavioral thermoregulation, evolutionary change, and/or phenotypic plasticity of the upper thermal limits (*Visser, 2008*).

Corresponding author
Luis E. Castañeda,
luis.castaneda@uchile.cl

Evolutionary adaptation to increasing environmental temperatures depends on genetic variation associated with upper thermal limits. While some studies indicate that heat tolerance has limited evolutionary potential for responding to global warming (*Chown et al., 2009*; *Mitchell & Hoffmann, 2010*; *Kellermann et al., 2012*), both theoretical and empirical evidence indicates that estimates of heat tolerance and its heritability depend on the methodological context. Specifically, organisms exposed to chronic thermal stress (*e.g.*, slow-ramping assays or static assays sublethal temperatures) have lower thermal resistance and heritability values than those exposed to acute thermal stress (*e.g.*, fast-ramping assays or static assays using extremely high temperatures) (*Chown et al., 2009*; *Mitchell & Hoffmann, 2010*; *Rezende, Tejedo & Santos, 2011*; *Blackburn et al., 2014*; *van Heerwaarden, Malmberg & Sgrò, 2016*; *Castañeda et al., 2019*; but see *Soto et al., 2025*). This methodological effect on heat tolerance–and its heritability–has been explained because thermal assays commonly are usually carried out by placing individuals in closed vials with no access to water and food. As the temperature increases, so do energy expenditure and the evaporative water loss. Therefore, individuals in the slow ramping rate should consume more energy and lose more water than those individuals in the fast ramping rate (*Rezende, Tejedo & Santos, 2011*; *Santos, Castañeda & Rezende, 2012*).

However, a recent study reported that *Drosophila subobscura* evolved higher heat tolerance regardless of the heating rate used during the artificial selection experiments (slow-ramping rate: 0.08 °C min$^{-1}$ and fast-ramping rate: 0.4 °C min$^{-1}$) (*Mesas, Jaramillo & Castañeda, 2021*). Artificial selection under laboratory conditions has a long history providing information on the adaptive evolution of specific selective scenarios (*Lenski & Bennett, 1993*; *Garland, 2003*; *Fuller, Baer & Travis, 2005*; *Gibbs & Gefen, 2009*). In particular, the experimental evolution of heat tolerance has been assessed in several species, including fish, corals, and insects (*Baer & Travis, 2000*; *Kelly, Sanford & Grosberg, 2012*; *Geerts et al., 2015*; *Esperk et al., 2016*). Experimental evolution of heat tolerance has also been studied in several *Drosophila* species, including *D. melanogaster* (*Gilchrist & Huey, 1999*; *Folk et al., 2006*), *D. subobscura* (*Quintana & Prevosti, 1990*; *Mesas, Jaramillo & Castañeda, 2021*; *Mesas & Castañeda, 2023*), and *D. buzzatti* (*Krebs & Loeschcke, 1996*). Interestingly, several of these experiments selecting for higher heat tolerance in *Drosophila* have found correlated responses in other traits such as starvation resistance, desiccation resistance, and heat shock proteins (*Hoffmann et al., 1997*; *Feder et al., 2002*; *Bubliy & Loeschcke, 2005*).

Artificial selection to increase heat tolerance of *D. subobscura* demonstrated that the correlated responses in thermal performance, metabolic flux, and fecundity depend on the ramping rate used. For instance, *Mesas, Jaramillo & Castañeda (2021)* revealed that lines selected with a fast ramping rate showed a higher optimum temperature and a narrower thermal amplitude on their thermal performance curves compared to control lines. Similarly, *Mesas & Castañeda (2023)* reported that selected lines with a slow ramping rate exhibited increased fecundity and reduced hexokinase activity (*i.e.*, an enzyme involved in the glucose metabolism) compared to control lines of *D. subobscura*; whereas, contrary to the expectations, there was no evolutionary change in metabolic rate in response to selection for increased heat tolerance in *D. subobscura* (*Mesas & Castañeda, 2023*). Thus,

there is evidence that the heat stress intensity during selection of heat tolerance determines the magnitude of the evolutionary responses of performance, metabolic, and life-history traits; however, its consequences on resistance to other stressors have not yet explored. Natural populations are regularly subjected to multiple environmental stressors, and it is well-established that enhanced tolerance to one stressor can enhance tolerance to another (*Rodgers & Gomez-Isaza, 2023*). These correlated responses among resistance traits are expected due to their shared physiological (*e.g.*, costly responses to stress) and/or a genetic basis (*e.g.*, linkage disequilibrium and pleiotropy) basis (*Bubliy & Loeschcke, 2005*; *Mauro & Ghalambor, 2020*). Thus, cross-tolerance induced by thermal stress has been widely studied in several arthropod species, increasing resistance to desiccation, insecticides, and pathogens (*Kalra, Tamang & Parkash, 2017*; *Rodgers & Gomez-Isaza, 2021*; *Singh, Arun Samant & Prasad, 2022*).

The present study investigates the effects of heat intensity selection for increasing knockdown temperature on the cross-tolerance evolution of resistance traits in *D. subobscura*: knockdown time at different stressful temperatures, thermal-death-time curves (TDT), desiccation resistance, and starvation resistance. The heat intensity selection was established by using two thermal selection protocols that differed in the rate of temperature increase to measure the heat knockdown temperature: selection using a slow ramping rate (0.08 °C min$^{-1}$, hereafter slow-ramping selection) and selection using a fast-ramping rate (0.4 °C min$^{-1}$, hereafter fast-ramping selection). The present work emphasizes the study of TDT curves, because they represent an integrative approach to evaluate the relationship between the probability of survival and the intensity and duration of heat stress, as they allow the estimation of the critical thermal maxima (CT$_{max}$) and thermal sensitivity using the thermal tolerance measurements obtained at different stress temperatures (*Rezende, Castañeda & Santos, 2014*). Here, it is expected that fast-ramping selected lines will evolve higher heat tolerance than slow-ramping selected lines (*e.g.*, higher knockdown time at highly stressful temperatures and higher CT$_{max}$) because fast-ramping protocols reduce the confounding effects (*e.g.*, hardening, rate of resource use) on heat tolerance associated with the assay length (see *Rezende, Tejedo & Santos, 2011*; *Santos, Castañeda & Rezende, 2012*; *Mesas, Jaramillo & Castañeda, 2021*). In contrast, slow-ramping selection lines should exhibit higher desiccation and starvation resistance because individuals with higher starvation and desiccation resistance exhibit higher thermal tolerance during long assays. This type of study will allow us to understand whether the correlated responses to selection for increased thermal tolerance depend on the environmental context, and thus to try to make better evolutionary predictions about the adaptive capacity of species in response to global warming.

## MATERIALS AND METHODS

Portions of this text were previously published as part of a preprint (*Castañeda, 2024*).

### Sampling and maintenance

*D. subobscura* females were collected in the spring 2014 at the Botanical Garden of the Universidad Austral de Chile (Valdivia, Chile; 39° 48′S, 73° 14′W) using plastic traps

containing banana/yeast baits. Two hundred females were collected and placed individually in plastic vials containing David's killed-yeast *Drosophila* medium to establish isofemale lines. In the next generation, 100 isofemale lines were randomly selected, and 10 females and 10 males per line were placed in an acrylic cage to establish a large, outbred population. In the next generation, the flies from this cage were randomly divided into three population cages (R1, R2, and R3), attempting to assign the same number of flies to each cage. After three generations, the flies in each replicate cage were randomly divided into four population cages, trying to assign the same number of flies to each cage. This procedure established 12 population cages, which were randomly assigned to each artificial selection protocol in triplicate: fast-ramping selection, fast-ramping control, slow-ramping selection, and slow-ramping control lines (Fig. S1). During the selection experiments, population cages were maintained at 18 °C (12:12 light-dark cycle, 50–60% humidity) in a discrete generation, controlled larval density regime (*Castañeda, Rezende & Santos, 2015*). Each population cage had a population size of 1,000–1,500 breeding adults.

## Heat tolerance selection

For each replicate cage, 120 four-day-old virgin females were randomly collected and placed individually in vials containing food medium. Two four-day-old males were randomly collected from each replicate cage and placed in the same vial as each female from the same replicate line. They were allowed to mate for 2 days, during which females laid eggs in the vial. Then, males were removed from the vials and females were placed individually in capped 5-mL glass vials. The vials containing eggs were placed in an incubator at 18 °C with a 12:12 light-dark cycle and 50–60% humidity. The offspring from these vials (hereafter referred to as "offspring vials") will be used to found the next generation of each replicate according to the selection protocol explained below.

The glass vials containing each mated female were attached to a plastic rack and immersed in a water tank (tank size: 35 cm × 42 cm × 45 cm; filling volume: 60 L) with an initial temperature of 28 °C (± 0.03 °C), controlled by a heating immersion unit (model ED; Julabo Labortechnik, Seelbach, Germany). After an equilibration period of 10 min, the temperature was increased to 0.08 °C min$^{-1}$ for the slow-ramping selection protocol or 0.4 °C min$^{-1}$ for the fast-ramping selection protocol. Assays were stopped when all female flies collapsed. Each assay was recorded using a high-resolution camera (model D5100; Nikon, Tokyo, Japan) and then visualized to score the knockdown temperature for each fly, defined as the temperature at which each fly ceased to show coordinated movements. After each assay females from each replicated line assigned to the selection protocols were ranked by knockdown temperature. Four virgin females from the offspring vials of the 40 flies with the highest knockdown temperature (top 30% of each assay) were selected at random to establish the next generation. For the fast and slow control lines, the knockdown temperature was measured as described above, but the progeny was randomly selected from the offspring vials to establish the next generation (*i.e.*, regardless of the knockdown temperature of their mother). This artificial selection experiment was performed for 16 generations, after which flies from each selection treatment were placed

in separate acrylic cages and maintained without selection (*e.g.*, relaxed selection) at 18 °C and a 12:12 light-dark cycle.

## Knockdown time in static assays

Eggs were collected from each population cage and transferred to vials at a density of 40 eggs/vial. At 4 days of age, ten females and ten males from each population cage were tested to measure their heat knockdown time at four different static temperatures: 35, 36, 37, and 38 °C. This experimental design allowed the measurement of 960 flies (10 flies × 2 sexes × 4 static temperatures × 4 selection treatments × 3 replicated lines). Static assays were performed similarly to knockdown temperature assays, but constant temperatures were used instead of ramping temperatures. A total of 240 flies were measured for each static temperature, except for the assay at 35 °C (178 flies) because two flies died before the start of the assay, and a video file of one assay was corrupted (data for 60 flies were lost). For the 37 °C assay, four flies died before the assay began, and the collapse time could not be measured for six flies. Finally, for the 38 °C assay, three flies died before the start of the assay and the collapse time could not be measured for five flies. Heat knockdown assays were performed in generation 23 (Fig. S1).

## Desiccation and starvation resistance

Eggs from each replicate cage were collected and placed in vials at a density of 40 eggs/vial. Only fast control lines were measured as control lines. This decision was based on logistical reasons (*i.e.*, the high number of vials) and statistical support because fast and slow control lines did not differ in their knockdown times and $CT_{max}$ values (see *the Results* section).

For desiccation resistance assays, flies were separated by sex. Five flies of the same sex were placed in a 15 mL plastic vial containing five $CaSO_4$-based desiccant droplets (Drierite®; W.A. Hammond Drierite Co., Xenia, OH, USA). The vials were sealed with parafilm to prevent water enters to vials (flies had no access to food or water during the assay). For starvation resistance assays, flies were separated by sex and the five flies of the same sex were placed in a 15 mL plastic vial with a mixture of agar and water (flies had access to water but no food). The vials containing flies were placed in an incubator connected to a PELT-5 thermal controller (Sable Systems International, Las Vegas, NV, USA), which allowed to maintain a constant temperature of 18 °C. A cold light connected to a timer was placed inside the incubator to maintain a 12L:12D photoperiod cycle throughout the experiments.

For both desiccation and starvation resistance assays, the number of live flies was counted every 3 h until all the flies were dead. Desiccation and starvation resistance were measured in 126 vials containing five flies each, respectively (7 vials × 2 sexes × 3 selection treatments × 3 replicate lines). These experiments were conducted using flies from generation 24 (Fig. S1).

## Statistical analysis

Normality and homoscedasticity were tested for all variables, and the knockdown times were squared root transformed to meet the parametric assumptions (*Quinn & Keough, 2002*). All analyses were performed with R software (*R Core Team, 2024*).

## Heat tolerance

The selection effect was tested separately for the fast- and slow-ramping datasets because the knockdown temperature is higher in fast-ramping assays than in slow-ramping assays (*Chown et al., 2009*; see *Mesas, Jaramillo & Castañeda, 2021*). For the knockdown time analysis, a mixed linear model with ramping selection (fixed effect with fast-control, slow-control, fast-selection, and slow-selection lines as levels), sex (fixed effect with females and males as levels), and replicate lines nested within the thermal selection (random effect with replicates 1, 2 and 3 as levels) was performed using the library *lme4* package for R (*Bates et al., 2015*). Fixed effects were tested using a type-III ANOVA, and the random effect was tested using a likelihood ratio test (LRT) by comparing the models that included or excluded replicate lines. Both tests were carried out using the library *lmerTest* package for R (*Kuznetsova, Brockhoff & Christensen, 2017*). If the selection effect was significant, the *emmeans* package for R (*Lenth, 2025*) was used to perform *posteriori* comparisons with a false-discovery rate (FDR) approach.

Knockdown times were also used to plot the survival curves based on the Kaplan-Meier formula using the *survfit* function implemented in the *survival* package for R (*Therneau, 2024*).

## Thermal death time curves (TDT)

Average knockdown times were calculated for each sex, replicate line, and selection treatment combination (Table S1). These values were regressed against the assay temperatures according to Eq. (1) (*Rezende, Castañeda & Santos, 2014*):

$$log_{10}t = \frac{CT_{max} - T}{Z}, \tag{1}$$

where $T$ is the assay static temperature (°C), $CT_{max}$ is the upper thermal limit (°C), $t$ is the knockdown time (min), and $z$ is the thermal sensitivity. These curves allowed the estimation of $CT_{max}$ as the extrapolated temperature that would result in a knockdown time of $log_{10} t = 0$ (*i.e.*, knockdown time at 1 min) and the estimation of the thermal sensitivity ($z = -1/slope$), where lower $z$ value, the higher the thermal sensitivity.

Using Eq. (1), 24 TDT curves (2 sexes × 3 replicate lines × 4 selection protocols) were fitted, from which $CT_{max}$ and $z$ values were estimated as described above. A linear model with ramping selection treatment (levels: fast-control, slow-control, fast-selection, and slow-selection lines), sex (levels: females and males), and their interaction was performed to evaluate their effects on $CT_{max}$ and $z$ values. TDT analysis did not include replicate lines as a random effect because only one $CT_{max}$ and $z$ value was estimated by each replicate line. If the selection treatments had a significant effect on $CT_{max}$ or $z$, an FDR approach was used to perform *posteriori* comparisons. Additionally, a linear mixed model with

ramping selection (fixed effect with fast-control, slow-control, fast-selection, and slow-selection lines as levels), sex (fixed effect with females and males as levels), and replicate lines nested within the thermal selection (random effect with replicates 1, 2 and 3 as levels), and assay temperatures (as covariate) was fitted on the knockdown time using the *lmer* package for R.

### Desiccation and starvation resistance

To determine the lethal time at which 50% of flies of each vial were dead ($LT_{50}$), a generalized linear model following a binomial distribution was fitted with the proportion of flies alive as the dependent variable and time as the predictor variable. The generalized linear model was run using the g*lm* function of the *lme4* package for R (*Bates et al., 2015*). The $LT_{50}$ of each vial was then estimated using the function *dose.p* from the *MASS* package for R (*Venables & Ripley, 2002*).

To estimate the median $LT_{50}$ and the 95% confidence intervals for each selection treatment and sex, $LT_{50}$ was transformed into a survival object using the *surv* and *survfit* functions of the *survival* package for R (*Therneau, 2024*). This procedure also allowed to estimate the survival curves in each vial. Finally, to test the effect of selection treatment (fixed effect with levels control, fast-selection, and slow-selection lines), replicate lines (random effect with levels R1, R2 and R3), and sex (fixed effect with levels females and males) on desiccation and starvation resistance, a mixed effect Cox proportional regression model was fitted with $LT_{50}$ as the dependent variable. The mixed effect Cox model was run using the *coxme* function of the *coxme* package (*Therneau, 2024*). To evaluate the variability among replicate lines, the same model was ran, excluding the replicate line effect, using the *cox* function of the *survival* package (*Therneau, 2024*). An LRT was used to compare mixed *vs.* fixed models.

## RESULTS

### Knockdown temperature evolution

Knockdown temperature evolved in response to artificial selection for increased heat tolerance, regardless of the ramping assay protocol. For the fast-ramping comparison, selected lines showed higher knockdown temperature than control lines (mean fast-ramping selection lines ± SD = 37.71 ± 0.68 °C and mean fast-ramping control lines ± SD = 37.23 ± 0.79 °C; $F_{1,4}$ = 32.0, $P$ = 0.005). For the slow-ramping comparison, selected lines showed higher knockdown temperature than control lines (mean slow-ramping selection lines ± SD = 35.48 ± 0.66 °C and mean fast-ramping control lines ± SD = 34.97 ± 0.72 °C; $F_{1,4}$ = 41.7, $P$ = 0.003). These results were previously reported by *Mesas, Jaramillo & Castañeda (2021)* and are reported here to show that selected lines used in this study evolved higher thermal tolerance compared to control lines.

### Knockdown time evolution

As expected, the knockdown time decreased significantly as the static temperatures increased ($F_{1,877}$ = 649.1, $P < 2 \times 10^{-16}$). The knockdown time (mean ± SE) for each static

**Table 1 Effect of selection, sex and interaction on knockdown time in *Drosophila* subobscura.**

| Knockdown time | Selection | Sex | Selection × Sex |
|---|---|---|---|
| Static assay at 35 °C | $F_{3,170} = 0.62$ | $F_{1,170} = 8.64$ | $F_{3,170} = 0.64$ |
| | $P = 0.60$ | **$P = 0.004$** | $P = 0.59$ |
| Static assay at 36 °C | $F_{3,232} = 9.86$ | $F_{1,232} = 2.65$ | $F_{3,232} = 0.74$ |
| | **$P = 3.8 \times 10^{-6}$** | $P = 0.10$ | $P = 0.53$ |
| Static assay at 37 °C | $F_{3,222} = 18.39$ | $F_{1,222} = 0.001$ | $F_{3,222} = 2.05$ |
| | **$P = 1.1 \times 10^{-10}$** | $P = 0.97$ | $P = 0.11$ |
| Static assay at 38 °C | $F_{3,224} = 1.93$ | $F_{1,224} = 4.63$ | $F_{3,224} = 2.44$ |
| | $P = 0.13$ | **$P = 0.032$** | $P = 0.07$ |

Note:
Mixed linear effect model for the knockdown time of *D. subobscura* assayed at four static temperature assays. For simplicity, results for the random effect (replicate lines) are not shown because they were not statistically significant (see Materials and Methods). Significant effects ($P < 0.05$) are indicated in boldface type.

assay are as follows: 35 °C = 33.77 ± 0.87 min; 36 °C = 16.98 ± 0.44 min; 37 °C = 8.82 ± 0.23 min; and 38 °C = 6.68 ± 0.18 min.

Knockdown times only differed significantly between selection treatments when flies were assayed at 36 and 37 °C (Table 1; Fig. 1). At these temperatures (Figs. 1C, 1E), slow-ramping selection lines showed higher knockdown time than slow-ramping control lines (36 °C: $P = 0.005$ and 37 °C: $P = 0.004$; Tables S1, S2). Also, fast-ramping selection lines showed higher knockdown time than fast-ramping control lines (36 °C: $P = 0.005$ and 37 °C: $P < 0.001$; Table S1, S2). In addition, fast-ramping selection lines showed higher knockdown time at 37 °C than slow-ramping selection lines ($P < 0.001$) but not at 36 °C ($P = 0.053$) (Table S1, S2; Figs. 1C, 1E), whereas fast-ramping control and slow-ramping control lines owed the similar knockdown time at all static temperatures (Table S2; Fig. 1). On the other hand, replicate lines showed similar knockdown time at all static temperatures (LRT: $\chi^2_1 = 0$, $P = 1$). In addition, females showed higher knockdown time than males but only when flies were assayed at 35 and 38 °C (Table 1; Figs. 1B, 1H). Finally, non-significant interactions between selection and sex were found for all static temperatures (Table 1).

## TDT curves evolution

Linear regressions between $\log_{10}(LT_{50})$ and assay temperatures enabled the estimation of 24 TDT curves (4 selection treatments × 3 replicate lines × 2 sexes) with high coefficients of determination (mean $R^2 = 0.946$, range: 0.820–0.989; Table S4), confirming that heat knockdown time is linearly related to stressful sublethal temperatures. From these TDT curves, the average $CT_{max}$ (mean ± SE) was 41.21 ± 0.02 °C, and the average $z$ (mean ± SE) was 4.18 ± 0.01 °C. A two-way ANOVA showed that a significant effect of selection treatments on $CT_{max}$ ($F_{3,16} = 5.16$, $P = 0.011$; Fig. 2A), but no effect of sex ($F_{1,16} = 0.008$, $P = 0.92$) or the interaction between selection and sex ($F_{3,16} = 2.21$, $P = 0.13$). *Posteriori* comparisons indicated that slow-ramping selection lines showed higher $CT_{max}$ than the slow-ramping control lines ($t_{16} = 2.86$, $P = 0.034$). In addition, fast-ramping selection lines

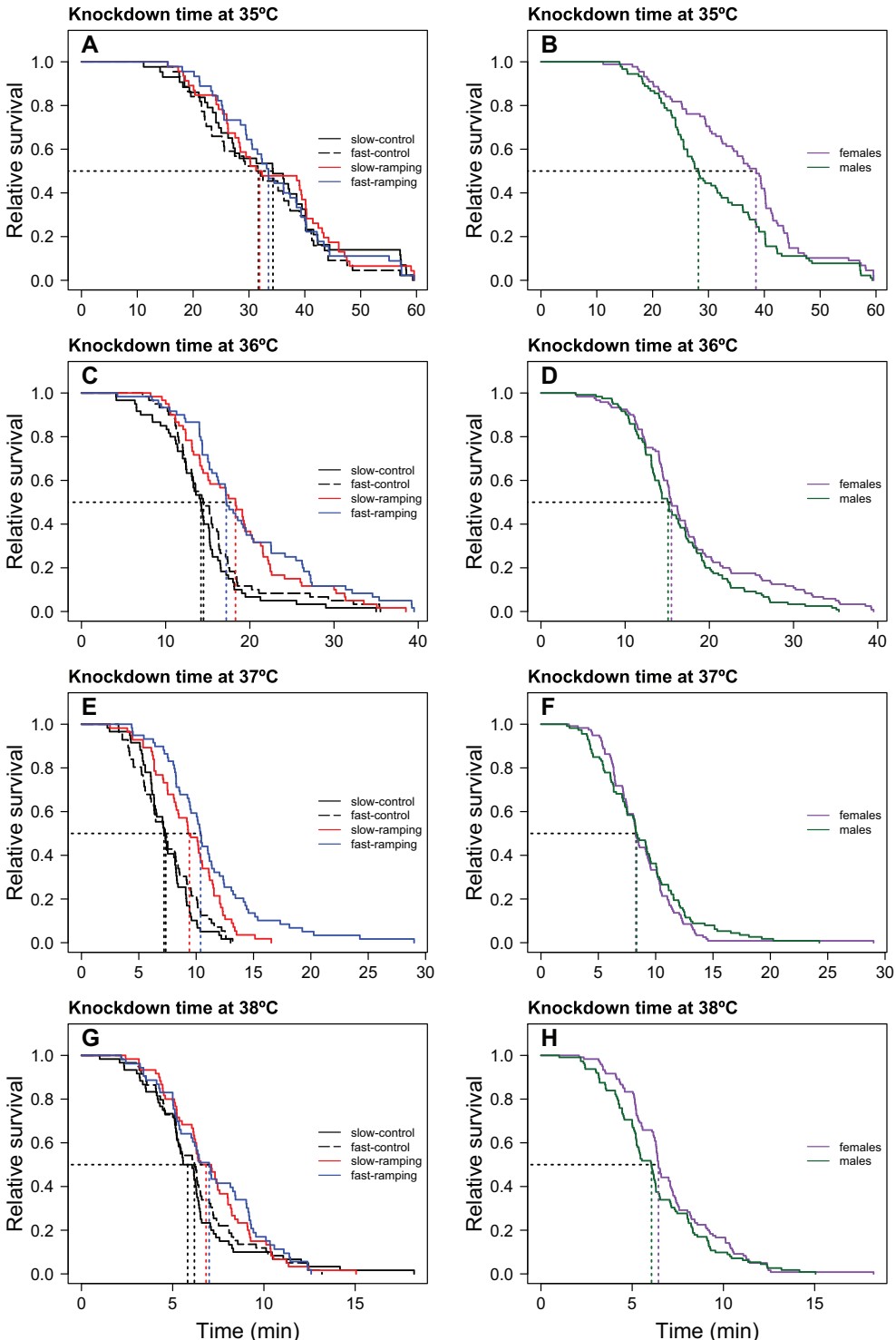

**Figure 1 Heat-induced mortality in *Drosophila subobscura* flies assayed at four static temperatures.**
Heat knockdown time measured at 35 (A), 36 (C), 37 (E), and 38 °C (G) of slow-ramping control (solid black line), fast-ramping control (dashed black line), slow-ramping selection (red line), and fast-ramping selection lines (blue lines). Heat knockdown time measured at 35 (B), 36 (D), 37 (F), and 38 °C (H) of female (purple line) and male (green line) flies. Dotted lines indicate the median knockdown time for each selection protocol (A, C, E and G) and sex (B, D, F and H).

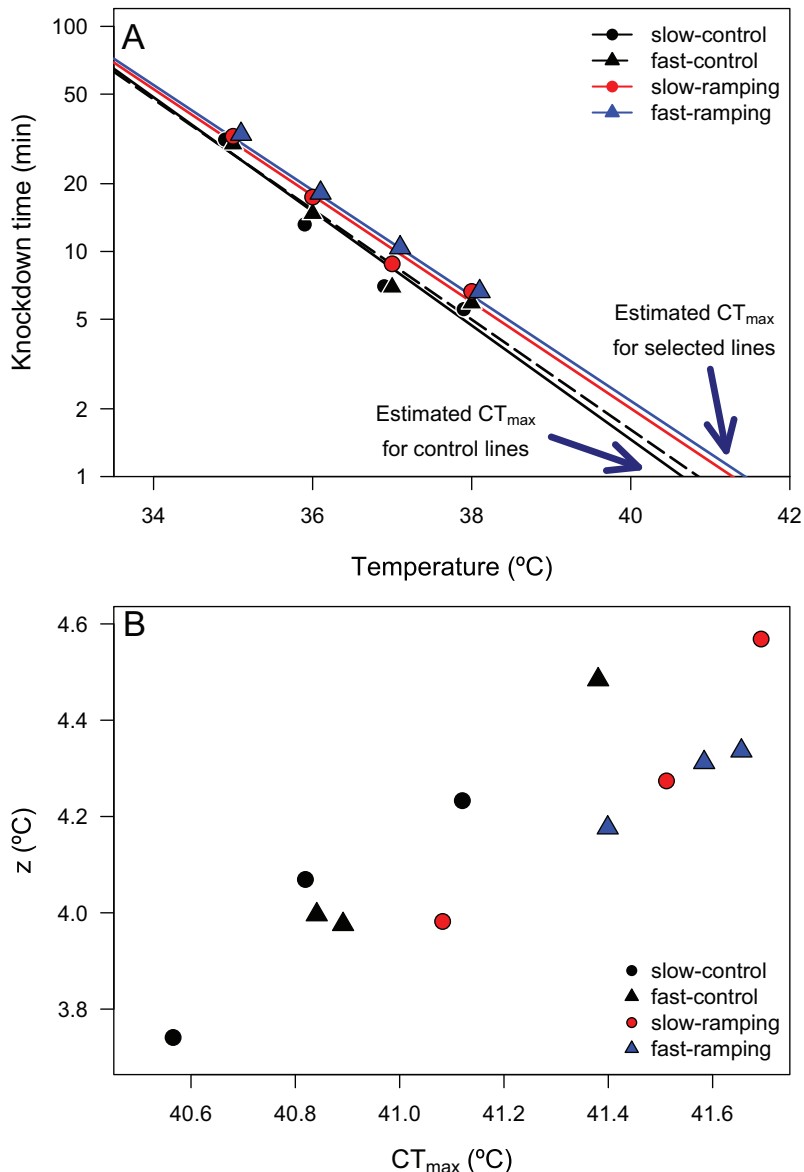

**Figure 2 Thermal death time curves (TDT) of *Drosophila subobscura*.** (A) Thermal death curves for control (black solid and dashed lines) and selected (red and blue lines) lines for increasing heat tolerance in *D. subobscura*. Symbols represent the average knockdown time at the different assay temperatures. Each symbol represents the average knockdown time for each replicate line for each thermal regime: slow-control (black circle), fast-control (black triangle), slow-ramping (red circle), and fast-ramping (blue triangle). (B) Relationship between $CT_{max}$ and z for slow-ramping control (solid black line), fast-ramping control (dashed black line), slow-ramping selection (red line), and fast-ramping selection lines (blue lines). Each symbol represents the $CT_{max}$ and z estimated for each replicate line.

also showed higher $CT_{max}$ than the fast-ramping control lines ($t_{16} = -2.49$, $P = 0.049$). While control lines of the fast-ramping and slow-ramping treatments showed similar $CT_{max}$ values ($t_{16} = 0.93$, $P = 0.44$). Regarding z (*i.e.*, thermal sensitivity; Fig. 2), it shows no significant effects of selection treatments ($F_{2,18} = 1.04$, $P = 0.37$), sex ($F_{1,18} = 1.27$, $P = 0.28$),
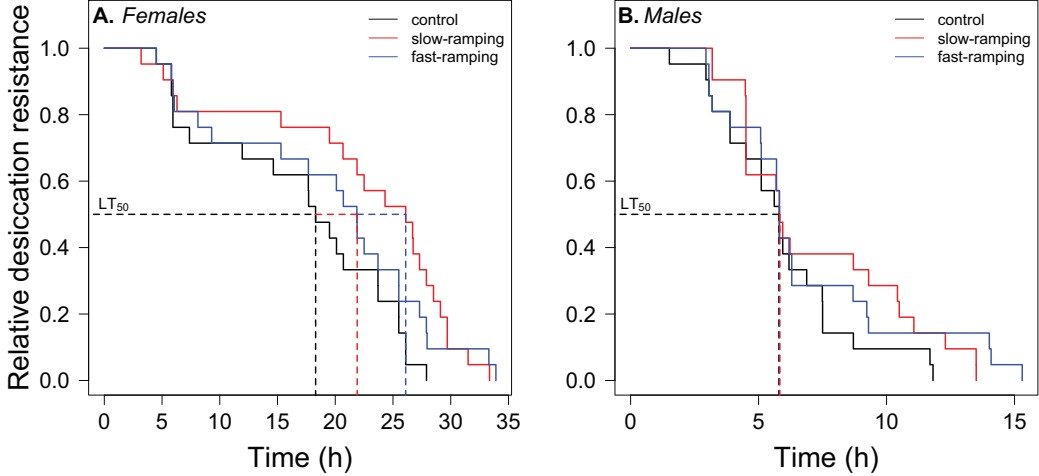

**Figure 3** **Correlated response of desiccation resistance in _Drosophila subobscura_.** Desiccation survival curves of (A) females and (B) males from control (black line), slow-ramping selection (red line), and fast-ramping selection lines (blue lines) of _D. subobscura_. Dashed lines indicate the median mortality time for each selection protocol (pooled replicate cages).

nor the interaction between selection treatments and sex ($F_{2,18} = 1.82$, $P = 0.19$). In summary, the evolution of a higher $CT_{max}$ is not associated with an evolutionary change in thermal sensitivity (Fig. 2B). In addition, the relationship between TDT parameters did not change with the evolutionary increase of $CT_{max}$ ($r_{control-lines} = 0.981$ and $r_{selection-lines} = 0.929$; Z-test = 1.06, $P = 0.29$).

## Desiccation resistance evolution

Survival analysis showed a significant effect of sex and selection treatment on desiccation resistance, but not an interaction between both effects (Table S5). Variation among replicate cages was not significant (LTR: $\chi^2_1 = 1.23$, $P = 0.27$). Males showed a higher risk of desiccation than female flies (hazard ratio = 7.11, $P < 2 \times 10^{-7}$; Fig. 3). Females of selected lines showed higher desiccation resistance than control lines (LTR: $\chi^2_2 = 6.72$, $P = 0.03$; Fig. 3A). Specifically, females of the slow-ramping selection lines showed a higher desiccation resistance than females of the control lines (hazard ratio = 0.42, $P = 0.009$), whereas females of the fast-ramping selection and control lines showed similar desiccation risk (hazard ratio = 0.56, $P = 0.072$). On the other hand, males showed no differences in desiccation resistance between selected and control lines (LTR: $\chi^2_2 = 1.88$, $P = 0.4$; Fig. 3B).

## Starvation resistance evolution

A significant effect of sex, selection treatment, and the interaction between the two effects on starvation resistance was found in the survival analysis (Table S6). Variation among replicate cages was not significant (LTR: $\chi^2_1 = 0.0034$, $P = 0.95$). Males had lower starvation resistance than female flies (hazard ratio = 22.75, $P < 1 \times 10^{-16}$; Fig. 4). In female flies (Fig. 4A), fast-ramping selection and slow-ramping selection lines showed a higher starvation risk than control lines (hazard ratio = 2.37, $P = 0.009$; and hazard ratio = 2.20,

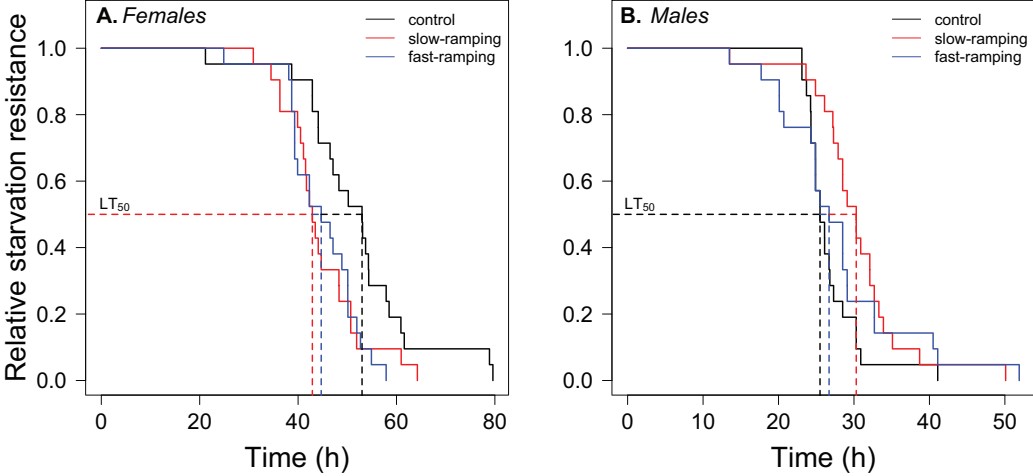

**Figure 4 Correlated response of starvation resistance in *Drosophila subobscura*.** Starvation survival curves of (A) females and (B) males from control (black line), slow-ramping selection (red line), and fast-ramping selection lines (blue lines) of *D. subobscura*. Dashed lines indicate the median mortality time for each selection protocol (pooled replicate cages).

$P = 0.014$, respectively). In contrast, male flies had an opposite pattern (Fig. 4B): slow-ramping selection lines had a lower starvation risk than control lines (hazard ratio = 0.50, $P = 0.03$), but nonsignificant differences were found between fast-ramping selection and control lines (hazard ratio = 0.64, $P = 0.16$).

# DISCUSSION

Studying the evolutionary responses of thermal limits is key to understanding the adaptive responses and evolutionary constraints to global warming. Cross-tolerance studies can provide valuable information on the evolutionary response to multiple environmental stressors. Cross-tolerance evolution has been reported among different resistance traits (*Hoffmann & Parsons, 1993*; *Bubliy & Loeschcke, 2005*; *Stazione et al., 2020*; *Singh, Arun Samant & Prasad, 2022*), but the magnitude of the evolutionary response could be explained by the trait under direct selection or the stress intensity (*Gerken, Mackay & Morgan, 2016*). Here, artificial selection for heat tolerance (*i.e.*, knockdown temperature) resulted in correlated responses in heat knockdown time, the thermal tolerance landscape (TDT curves), desiccation resistance, and starvation resistance. However, these responses depended on the intensity of thermal selection and sex, suggesting that predictions of evolutionary responses to global change should depend on the interaction among abiotic and biotic factors (*Urban et al., 2024*).

Different approaches to measuring the upper thermal limit of ectotherms produce different genetic and phenotypic estimates. Although slow ramping assays have been recommended to mimic daily temperature changes experienced by organisms in natural scenarios (*Overgaard, Kristensen & Sørensen, 2012*), fast-ramping assays produce more reliable estimates of upper thermal limits and heritabilities (*Rezende, Tejedo & Santos, 2011*; *Castañeda et al., 2019*). Consequently, different ramping assays used during

selection experiments should have important consequences for the evolutionary response of heat tolerance. However, *Mesas, Jaramillo & Castañeda (2021)* found that the evolutionary response of heat tolerance was independent of the ramping rate in *D. subobscura*, but the correlated responses of the thermal performance curves (*Mesas, Jaramillo & Castañeda, 2021*) or the energy metabolism depended on the intensity of the thermal selection (*Mesas & Castañeda, 2023*).

In the present study, the evolution of knockdown temperature (*i.e.*, heat tolerance measured in dynamic assays) induced a correlated response on the heat knockdown time (*i.e.*, heat tolerance measured in static assays) when it was assayed at intermediate temperatures (36 and 37 °C), but not at the lowest or highest temperatures (35 °C and 38 °C) assayed in this study. These findings can be explained by the fact that thermal tolerance at 35 °C may depend on the physiological state of the organism during prolonged thermal assays (*e.g.*, energy resource availability; see *Rezende, Tejedo & Santos, 2011*, but also see *Overgaard, Kristensen & Sørensen, 2012*), blurring the evolutionary response of heat tolerance, whereas the evolutionary response of heat tolerance at 38 °C may be limited by the physical properties of ectotherms (*e.g.*, protein denaturation, membrane permeability), which may not exhibit variability in the population examined in this study. However, *Castañeda, Rezende & Santos (2015)* evaluated the variability in heat tolerance among *D. subobscura* populations spanning a 1,000 km latitudinal range in a common-garden experiment. They found a clinal pattern when heat tolerance was assayed at 38 °C but not when ramping assays were used. However, comparisons of heat tolerance responses to natural and artificial selection should be taken with caution because the differences between the findings of *Castañeda, Rezende & Santos (2015)* and those shown here could be explained by the number of generations under thermal selection. According to *Begon (1976)*, *D. subobscura* can have between four and six generations per year, which makes it possible to estimate about 125 generations of selection since the introduction of *D. subobcura* in Chile until the study by *Castañeda, Rezende & Santos (2015)*.

In general, our findings support the idea that (1) the use of a single static temperature would miss genetic or phenotypic difference on heat tolerance, and (2) unifying several knockdown time estimates into a single approach (TDT curves) should be necessary to elucidate genetic and phenotypic patterns of heat tolerance in ectotherms (*Rezende, Castañeda & Santos, 2014*; *Jørgensen et al., 2021*). Specifically, TDT curves evolved in response to heat tolerance selection in *D. subobscura*, with fast- and slow-ramping selection lines evolved higher $CT_{max}$ in comparison to control lines ($\Delta CT_{max} = 0.49$ °C). This $CT_{max}$ differential value is slightly lower than reported by *Castañeda, Rezende & Santos (2015)*, who conducted a common garden experiment to determine the genetic differentiation in heat tolerance among *Drosophila subobscura* populations distributed along a latitudinal gradient in Chile (acclimated pooled $\Delta CT_{max} = 0.9$ °C). It is also lower than the $CT_{max}$ variation reported among *Drosophila* species (*Jørgensen, Malte & Overgaard, 2019*; *Alruiz et al., 2022*). Although the magnitude of $CT_{max}$ evolution is smaller than the variation at intra-and interspecific levels, it can still have important implications for survival in nature. For example, *Rezende et al. (2020)* estimated the TDT

curves of *D. subobscura* flies acclimated to cold and warm temperatures using data from *Castañeda, Rezende & Santos (2015)*. With differences of 0.8 °C for $CT_{max}$ and 1.2 °C for *z*, *Rezende et al. (2020)* predicted that cold-acclimated flies would experience high mortality in mid-spring and warm-acclimated flies would experience high mortality in mid-summer. On the other hand, although $CT_{max}$ and z (*i.e.*, thermal sensitivity) are phenotypically correlated (see *Castañeda, Rezende & Santos, 2015*; *Molina et al., 2023*), the evolutionary increase in $CT_{max}$ was not associated with a correlated response in thermal sensitivity (*z*). This result suggests that both thermal parameters are not genetically constrained, but further evidence from quantitative genetic studies is needed to assess the genetic association between $CT_{max}$ and *z*. A caveat for this finding could be related to the fact that thermal selection for heat tolerance was carried out over 16 generations, followed by seven generations of relaxed selection (*i.e.*, no selection). However, previous evidence suggests that differences in heat tolerance between control and selected lines were consistent between generations 16 and 25 (*Mesas, Jaramillo & Castañeda, 2021*). Indeed, *Passananti, Beckman & Rose (2004)* also reported that phenotypic values did not change after 35 generations of relaxed selection in desiccation-selected populations of *D. melanogaster*.

A working hypothesis of this study was that flies selected for higher heat tolerance using slow-ramping rate protocols would exhibit greater desiccation and starvation resistance than flies selected using fast-ramping selection protocols. This hypothesis is based on that flies assayed for heat tolerance in long assays are also exposed to desiccation and starvation stress during measurement trails (*Santos, Castañeda & Rezende, 2012*). According to the results reported here, this study provides partial support for this hypothesis. First, slow-ramping selection lines evolved a higher desiccation resistance in comparison to control lines and fast-ramping selection lines. However, this finding was only observed in female flies, while males of the different selection treatments did not show any difference in desiccation resistance. On the other hand, starvation resistance evolved in opposite directions depending on sex: females of the fast-ramping and slow-ramping selection lines showed lower starvation resistance than females of the control lines, whereas males of the slow-ramping selection lines showed higher starvation resistance than males of the control lines and fast-ramping selection lines. Differential evolutionary responses between the sexes could be due to heat thermal selection only being applied to females, which could have exacerbated the evolutionary responses of female flies. However, previous studies that artificially selected exaggerated male traits also found fitness consequences in females (*Harano et al., 2010*). Differential evolutionary responses between females and males can then be explained by sexually antagonistic selection on genetically correlated traits (*Eyer, Blumenfeld & Vargo, 2019*; *Fanara et al., 2023*). *Kwan et al. (2008)* reported that desiccation-selected females had higher desiccation resistance than desiccation-selected males, which can be explained by males using resources at a faster rate than females (*e.g.*, males lose weight, water, and metabolites faster than females). Sexual dimorphism in stress resistance traits has been mainly explained by differences in cuticular composition, resource storage, and energy conservation between the sexes (*Schwasinger-Schmidt,*

*Kachman & Harshman, 2012*; *Rusuwa et al., 2022*). Although energy content was not measured here, *Mesas & Castañeda (2023)* found that females of *D. subobscura* showed similar body mass and metabolic rate between control and heat-tolerance selected lines, suggesting that the energy metabolism cannot explain the correlated response for stress resistance traits. However, the same study found that heat-tolerance selected lines had higher fecundity than control lines, whereas previous studies have found negative associations between fecundity and starvation resistance in *D. melanogaster* (*Bubliy & Loeschcke, 2005*; *Kalra, Tamang & Parkash, 2017*). Then, the decrease in starvation resistance in females of the heat-selected lines could be related to increased fecundity, which is consistent with the reported trade-off between stress resistance traits and life-history traits (*van Noordwijk & de Jong, 1986*; *Rion & Kawecki, 2007*).

In conclusion, the present study shows that heat tolerance evolution is associated with evolutionary responses in other stress resistance traits, which may be explained by pleiotropic effects or linkage disequilibrium among the traits evaluated. However, further evidence through quantitative genetic or genomic studies are needed to elucidate the genetic basis of the cross-tolerance evolution in *D. subobscura*. In addition, this study provides evidence for fast evolutionary responses in ectotherms mediated by thermal selection, but the evolutionary outcomes depend on the intensity of the thermal stress (*Mesas & Castañeda, 2023*) and sex (*Rogell et al., 2014*; *Rusuwa et al., 2022*). This study also highlights the importance of *D. subobscura* as a suitable model to study thermal adaptation mediated by natural selection (*Huey, 2000*; *Gilchrist et al., 2008*; *Castañeda et al., 2013*; *Castañeda, Rezende & Santos, 2015*), and laboratory selection (*Santos et al., 2005*, *2021*; *Simões et al., 2017*; *Mesas, Jaramillo & Castañeda, 2021*; *Mesas & Castañeda, 2023*). In addition, this study highlights the relevance of experimental evolutionary studies for understanding the adaptive responses to climate change (*Mitchell & Whitney, 2018*; *Brennan et al., 2022*; *Kelly, 2022*). Finally, these results suggest that ectotherms may evolve in response to climate warming, but evolutionary responses may differ between sexes and/ or the warming rates experienced by natural populations, which may make it difficult to propose general trends in the fate of ectotherms in a changing world where temperature is not the only driver of climate change, but species are also expected to be exposed to changes in precipitation patterns and food availability.

## ACKNOWLEDGEMENTS

I thank Andres Mesas, Angélica Jaramillo, Julio Figueroa, and Jaiber Solano for their help with fly maintenance, experiments, and data management. Thanks to Pedro Simões, Inès Fragata, Marina Stamenkovic-Radak and one anonymous reviewer for their comments and suggestions during this review process in Peers Community In (PCI). I would also like to thank the editor and the three reviewers who participated in the PeerJ review process.

### Funding

This work was funded by the Fondo Nacional de Desarrollo Científico y Tecnológico (FONDECYT) [grant number 1140066]–Chile. Currently, Luis E. Castañeda is supported by the Agencia Nacional de Invertigación y Desarrollo (ANID) through the grants Anillo ATE230025, FOVI220194, and FOVI 230149. The funders had no role in study design, data collection and analysis, decision to publish, or preparation of the manuscript.

### Grant Disclosures

The following grant information was disclosed by the authors:
Fondo Nacional de Desarrollo Científico y Tecnológico (FONDECYT): 1140066.
Agencia Nacional de Invertigación y Desarrollo (ANID): ATE230025, FOVI220194, FOVI 230149.

### Competing Interests

The author declares that he has no competing interests.

### Author Contributions

- Luis E. Castañeda conceived and designed the experiments, performed the experiments, analyzed the data, prepared figures and/or tables, authored or reviewed drafts of the article, and approved the final draft.

### Data Availability

The data and scripts are available at Figshare: Castañeda, Luis (2023). Cross-tolerance evolution is driven by selection on heat tolerance in *Drosophila subobscura*. figshare. Dataset. https://doi.org/10.6084/m9.figshare.24085107.v6.

### Supplemental Information

Supplemental information for this article can be found online at http://dx.doi.org/10.7717/peerj.19743#supplemental-information.

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
