# Peer review of "Cross-tolerance evolution is driven by selection on heat tolerance in Drosophila subobscura"

_PeerJ, doi:10.7717/peerj.19743_

## Round 0.1 · original submission · Major Revisions

Dear Dr. Castañeda,

After this first review round, both reviewers believe your manuscript has merits and may be worthy of publication after reviews are implemented. Please check all the issues they raised and the best way to implement the suggested changes. As soon as the new version of your text is delivered, I am certain both reviewers will be available to reassess it.

Sincerely,

Daniel Silva

Reviewer 1 ·

Basic reporting

Luis E. Castañeda presents a well-structured experimental design to investigate correlated responses in heat tolerance, desiccation resistance, and starvation resistance in Drosophila subobscura. The study is grounded in a particularly compelling idea: under natural conditions, stressful temperatures rarely lead to organismal collapse through isolated physiological mechanisms. Instead, thermal stress interacts with other environmental factors, jointly determining survival. It is particularly interesting that, as organisms adapt to increased thermal tolerance, they may also gain resistance to other stressors that commonly interact with heat stress, such as desiccation and starvation. In this study, despite the signal is not particularly strong, results suggest that evolutionary responses to warming may differ between sexes (and depending on the rate of warming).

Overall, the manuscript is well written, and the author effectively guides the reader through a complex experimental framework and a substantial body of results. The topic is within the scope of PeerJ, and the research question is relevant in the current context of global change. Therefore, the study has great potential interest for researchers across a range of disciplines. For these reasons, I am supportive of the publication of the manuscript. Below, I outline a few specific points that require attention.

Experimental design

NO COMMENTS

Validity of the findings

NO COMMENTS

Additional comments

L90. Remove the stop after “of”.

L54-56. I am not a native English speaker, but the construction here seems odd. Starting a sentence with “however”, and connecting the next one with “while” makes the transition is not correct.

L65. Could the author provide a brief explanation of the physiological mechanisms behind this? (water loss by evapotranspiration?...).

L67-97. The author pro vides a long list of references here with a variety of practical examples. However, the paragraph can be strengthened by including a brief conceptual synthesis explaining why a correlated response to thermal selection is expected. For example, is there any common physiological mechanisms that behind this correlation?

L84-89. This sentence is too long (6 lines) and stops reading flow. I would suggest splitting it.

L98. The term “heat intensity selection” should be briefly defined.

L116. I missed a closing sentence at the end of the discussion giving some rationale, impact or justification to the study (like at the end of the abstract).

L254. No references in the “Results” section. This belongs to discussion.

289–290. “Then, control lines were pooled to increase the power to detect differences between selected and control lines”. – The analysis was presented as a full factorial design comparing four treatments (slow-ramping selection, slow-ramping control, fast-ramping selection, and fast-ramping control). However, in this sentence, the two control lines (slow and fast) are pooled to increase statistical power. This may be problematic because the initial analytical framework treats the control groups as different, implying potential differences between them. Pooling them post hoc without explicitly testing for or reporting a lack of significant differences may obscure meaningful variation. If the two control groups differ even slightly in thermal performance, combining them could distort the comparison with the selected lines. This should be justified with statistical evidence.

L335-337. The author stated that the evolution of heat tolerance confers "partial tolerance" to other stressors, but this point requires greater precision. This should be toned down. A consistent pattern was not observed across both sexes or selection treatments, so referring to "partial tolerance" may be an overgeneralization.

L338-370. This paragraph is extensive, containing several ideas. I suggest to split in two.

L338-370. The broader ecological relevance of fast vs. slow ramping assays is not sufficiently discussed. It would be interesting to elaborate a bit more on which assay type better approximates natural thermal stress scenarios.

L338-370. The claim that knockdown time only responded at intermediate temperatures (36 and 37 °C) is very interesting, but I think it is a bit underdeveloped.

L373. The biological relevance of the increase in CTmax (~0.49 °C) is not discussed. The author could improve this section by briefly discussing on whether this magnitude is expected to translate into meaningful differences in performance or survival under natural heat stress scenarios.

L375. It would be helpful to clarify whether the latitudinal differences in CTmax reported in Castañeda et al. (2015) can caused by plasticity, genetic divergence, or both. Otherwise, this comparison seems to mix fundamentally different processes.

L422. The suggestion that correlated responses are due to pleiotropy is speculative without genetic data. While this is acknowledged, a more cautious formulation (e.g. "may be explained by") is needed.

L425. The study involves 16 generations under selection, which is moderately fast in evolutionary terms, not necessarily "rapid". It may be more accurate to specify that responses occurred over a relatively short number of generations under artificial selection, instead of rapid adaptation under natural conditions.

Reviewer 2 ·

Basic reporting

The author of this study explores a crucial aspect of understanding global warming responses in an ectotherm, further testing the responses of cross-tolerance (desiccation and starvation trials), employing thermal death time modelling to further unravel the correlated responses to thermal stress (slow/fast ramping).

Experimental design

Study model is a fly and that has been historically used for similar work, so plenty is known, and the study design is replicated, conducted well and tested with working hypotheses in mind. I do believe that these types of studies are needed if we are to understand the global responses to complexities arising from climate change.

Validity of the findings

Overall, I believe that this study will be received well with a broad readership and future work can build upon these findings and the generality of these findings across other taxa.

Additional comments

Having said this, I did have some specific comments and clarification in the different sections, please see my specific comments (below). Hope these are clear and will help streamline the narrative generally.

L28-29: under selection of heat tolerance is missing.
L57-58; 63 and elsewhere throughout your ms, change to: methodological context
L58-61: here, under chronic stress, slow ramping etc. i would have thought that they have enough time to harden and build up resistance, although you claim the opposite?
L64-66: if chronic exposures are for longer and some survival is possible, there could be selection on dispersal- which might in turn provide hardened/acclimatised individuals some benefit during future chronic exposures, essentially becoming selected for dispersers (at least in those that can remain mobile and track climates)...
L82-84: lines of what? missing details..
L84-89: I am not 100% sure these sentences and pieces of evidence are going the direction it is intended to...in the first part you talk about increase heat tolerance and the enzymes responsible and in the second part you talk about increased fecundity and then? what enzymes were involved? Or does this have other interpretation altogether, please could you clarify?
L90: selected lines – do you mean “in the slow ramping rate selected lines? can you check this?”
L90-93: in the species of interest or any species, needs to be more specific here and clarify this at the beginning of the sentence itself!
L128-129: random assigning? Clarify
L131-132 & 153-154: please could you state the humidity maintained?
L137: again, here it is crucial to mention the exact background of the sexes. Females were paired with males (age matched?) of the same replicate line or some other line/stock? AND was each female mated with two, independent males?
L139: males were discarded- would suggest rewording this to "males were removed to stop mating" or something along these lines, as discarded seems pretty abrupt!
L139-140: what is the size of this water tank? how much water was used to top up the water bath?
L140-141: what is the accuracy pf this heating unit? temperature was measured at 28°C +/-0.5°C or 1°C. This is important for the reproducibility of your method.
L141-142: so, in essence, mated females were exposed here- correct?
L143: collapsed- what do you mean exactly by collapsed? Is it that where flies did not show coordinated movement and were not able to correct their orientation? Is this what you mean?
L146-148: at some point after this step, did you undertake any anonymisation of the treatments so that you controlled for any unconscious biases?
L162: but static- already said at the beginning of the sentence, so can be deleted
L171: was collected?
L176: Drierite- what is this, how was it sourced?
L176-177 & 177-178 (placing each sex in a separate..): where was this test conducted in a CT room or a chamber etc. Not clear, what were the conditions of the surroundings?
L182: what about the other characteristics like photoperiod, humidity, groups were mixed sexes or single sex groups?

L186: square root transformed- relevant reference for this method?
L190-192: could you simplify this by re-writing this sentence?
L198-199: this needs re-writing....i can see what you are trying to say here.
L206: replicate lines = replicate line?
L216: delete ‘a’ & 221: delete ‘curve’
L237: already mentioned each, can delete this 'each'
L239: curves in each vial- “but this is not your objective, you wanted to find out the effect of treatment by sex?”
Results section needs to be presented in such a way that it can be easily followed. At the moment to someone else reading the results, it comes across a bit too wordy, please could you write in the order you have tested your hypothesis with a clear objective and then going into the different aspects of the results! (just a suggestion).

Reviewer 3 ·

Basic reporting

.

Experimental design

.

Validity of the findings

.

Additional comments

This manuscript presents a well-executed and insightful investigation into the evolution of cross-tolerance, focusing on how selection for heat tolerance in Drosophila subobscura influences tolerance to additional environmental stressors. The authors offer a comprehensive and methodologically sound study that significantly contributes to our understanding of adaptive responses and the broader mechanisms of evolutionary resilience under multiple stress conditions.

The research is scientifically rigorous, methodologically robust, and addresses a question of substantial relevance within evolutionary biology. The manuscript is generally well-written and clearly organized, facilitating effective communication of the findings.

I recommend acceptance of this manuscript, pending minor revisions.

Minor Comments:

- It is unclear why the authors did not allow for a sufficient number of generations to stabilize the genome or gene pool. A period of 3–4 generations may be insufficient for genomic homogenization. Did you observe high variability across replicate populations? Have you considered examining life-history traits such as longevity or developmental time?

- The observed sexual dimorphism in desiccation resistance—evident in females but not males—raises intriguing questions. Why do you think this response evolved only in females? Could it be that the selection pressure is acting more strongly on females? Additionally, have you investigated whether this selection pressure influences reproductive traits?

---

## Round 0.2 · accepted · Accept

Dear Dr. Castañeda,

After this new review round, the reviewer believes the text is suitable for publication. Therefore, I am pleased to accept it for publication in PeerJ! Congratulations!

Best regards,
Daniel Silva

Reviewer 1 ·

Basic reporting

Here I review this manuscript for the second time. In my first review I was already very positive of the publication of this study. I proposed several minor issues and suggestions during my first review, and the author has answered or incorporated all of them. In this round, I have also checked the comments from the other reviewers (very positive in general), and I could see the author also have answered all of them very positively. After this second round of review, and based on my expertise and background, I have not identified any remaining issues that require further attention.

Experimental design

The study presents a highly elaborate experimental design aimed at investigating the interaction of multiple stressors with thermal tolerance. Specifically, it examines how tolerance varies depending on the rate of heating to which experimental individuals are subjected, as well as sex and other factors such as food availability and desiccation. The experimental design is complex and involves several levels of control. Moreover, the author works with both adults and eggs, and analyses various responses. While it is true that some molecular assays could have shed light on the mechanisms underlying the sex-specific responses or the evolutionary processes involved, the large number of individuals analysed and the relatively high level of replication are highly commendable (especially considering the technical challenges of separating individuals, measuring tolerance at the individual level, maintaining cultures, and so on). In general terms, the experiment is well presented and explained.

Validity of the findings

As I mentioned in my previous review, the validity of the findings remains meaningful, even though the observed signals are not particularly strong. However, we must keep in mind that working in a laboratory setting limits the ability to obtain results at ecological scales. I am convinced that the signal detected (the differences in thermal tolerance between sexes and under starvation or desiccation treatments) would likely become more pronounced if the number of generations were increased, more closely resembling the timescales over which evolutionary processes operate in nature.

In ecology and environmental sciences in general, bridging the gap between laboratory results and field-relevant mechanisms and predictions is a key challenge, especially if we aim to understand processes occurring under the accelerated pace of climate change. This study makes a commendable effort in that direction by using physiological experiments to address ecological questions. Furthermore, the conceptual framework employed of the thermal tolerance landscapes serves as a unifying approach to studying thermal responses in ectotherms.

Additional comments

I do not have any additional comment at this stage.

Reviewer 2 ·

Basic reporting

I am happy with the way the author has addressed the comments to the previous round of peer-review. I would like to congratulate the author on their efforts.

Experimental design

Reads well and is improved with clarity in the text.

Validity of the findings

Yes, this section too reads well and is much improved from the previous version.

Additional comments

I would like to congratulate the author on their efforts to make this study clearer and I don't have any further comments at this stage.